# Immediate and 15-Week Correlates of Individual Commitment to a “Green Monday” National Campaign Fostering Weekly Substitution of Meat and Fish by Other Nutrients

**DOI:** 10.3390/nu11071694

**Published:** 2019-07-23

**Authors:** Laurent Bègue, Nicolas Treich

**Affiliations:** 1LIP/PC2S, Univ. Grenoble-Alpes, France & MSH Alpes, CNRS, BP 47, CEDEX 9, 38040 Grenoble, France; 2Toulouse School of Economics, INRA, Université Toulouse Capitole, 21, All de Brienne, 31000 Toulouse, France

**Keywords:** meatless Monday, public health, sustainability, animal welfare, openness, vegetarianism, national campaign, food transition

## Abstract

Promoting healthier and more sustainable diets by decreasing meat consumption represents a significant challenge in the Anthropocene epoch. However, data are scarce regarding the effects of nationwide meat reduction campaigns. We described and analyzed the correlates of a national campaign in France (called “Green Monday”, GM) promoting the weekly substitution of meat and fish by other nutrients. Two cross-sectional online surveys were compared: a National Comparison sample (NC) of the French general population and a self-selected sample of participants who registered for the Green Monday campaign. A follow-up study was carried out in the GM sample, in which participants were asked during 15 weeks whether or not they had substituted meat and fish. There were 2005 participants aged 18–95 (47.7% females) in the NC sample and 24,507 participants aged 18–95 (77.5% females) in the GM sample. One month after the beginning of the campaign, 51.2% of the respondents reported they had heard about Green Monday in the NC sample, and 10.5% indicated they had already started to apply Green Monday. Logistic regression analysis showed that compared to the NC sample, participants belonging to the GM sample displayed a higher rate of females, Odds Ratio (OR) = 4.26, 95% Confidence Interval (CI): 3.86–4.71, were more educated, OR = 1.32, 95% CI: 1.28–1.36, had higher self-rated affluence, OR = 1.50, 95% CI: 1.42–1.58 and the size of their vegetarian network was greater, OR = 1.50, 95% CI: 1.41–1.58. They reported a slightly higher frequency of meat consumption, OR = 1.05, 95% CI: 1.01–1.10, while their frequency of fish consumption was lower, OR = 0.81, 95% CI: 0.76–0.87. Finally, the personality dimension Openness was more strongly endorsed by participants in the GM sample, OR = 1.79, 95% CI: 1.65–1.93. A multiple regression analysis indicated that Openness also predicted the number of participation weeks in the GM Sample (beta = 0.03, *p* < 0.009). In conclusion, specific demographic and personality profiles were more responsive to the national campaign, which could inform and help to shape future actions aiming at changing food habits.

## 1. Introduction

Food has a major impact on human health and the environment in the Anthropocene epoch [1]. In particular, the environmental footprint of meat production and the adverse health consequences of excessive meat consumption, such as risks of cancer [2], heart diseases [3], strokes [4,5,6], type 2 diabetes [5], and obesity [7,8] are now well-documented [9,10,11,12]. A recent meta-analysis study computed suboptimal diet based on 15 mortality and morbidity risk factors and recommended to limit the intake of red meat and processed meat in Western Europe and to increase the consumption of vegetables and whole grains [13]. Albeit less documented, some other benefits of an increase of vegetable consumptions have also been mentioned regarding gastroesophageal reflux disease [14] and asthma [15]. Moreover, the widespread non-therapeutic use of antibiotics in industrial animal food production increases the risks of developing antibiotic-resistant bacteria [16]. Meat production also causes contamination of water, air and soil and a global loss of biodiversity [16,17,18]. Some have also underlined the net loss of food resources: plant-based replacements could produce twofold to 20-fold more nutritionally similar food per unit of cropland [19]. It is estimated that cattle are responsible for 14.5% of worldwide global greenhouse gas emission [20,21,22].

In France, almost 70% of the adult population (18–79 years old) consume meat (poultry excluded). Current meat consumption of 47 g/day is consistent with the national recommendation, namely, no more than 70g/day according to the French reference agency ANSES [23]. However, there is a large heterogeneity in meat consumption since the standard deviation is 55 g/day [24], implying that an important proportion of meat consumers eat more than the maximal recommended level. Regarding processed meat, more than half of the adults eat more than the recommended maximal level by ANSES in France (i.e., 25 g/day). At a worldwide level, the global average per capita consumption of meat is still increasing, especially poultry meat, driven by population growth and the rise of average individual income [25]. Worldwide meat consumption has greatly increased in the last 60 years, with per capita consumption per year more than doubling: it raised from 17 to 43 kilos [26] and is projected to grow by 70% by 2050 [21]. The adoption of more climate-friendly food choices by the public represents an essential measure for reducing greenhouse gas emissions [27,28]. Available public interventions to stimulate change include taxation, nutritional labeling, as well as certification programs. However, in the food domain, fiscal or informational instruments often work imperfectly [29,30] and may be difficult to implement for economic and political reasons [31]. In developed countries, while climate regulation applies to all major polluting economic sectors, such as transport or energy, no government has yet implemented a carbon tax on meat, for instance.

Using bottom-up behavioral instruments to shape consumer food demand on a large scale represents another option. In order to foster a reduction of meat consumption, campaigns aiming at raising awareness of environmental and health benefits have been introduced in many countries [32]. Launched in 2003 by the Johns Hopkins Bloomberg School of Public Health, the “Meatless Monday” campaign is the most known initiative worldwide [33,34]. According to its website, it is currently active in about 40 countries [35]. A myriad of independent meatless day initiatives exist nowadays in schools, universities, hospitals and firms all around the world. Some have argued that meatless day campaigns are among the most effective strategies to reduce meat consumption worldwide [36]. These campaigns use a “foot-in-the-door” psychological technique, encouraging people to adopt a small commitment first, possibly inducing a larger commitment later. They can educate the public through active participation and in turn challenge dietary habits and routines. Furthermore, they can help address practical coordination problems among consumers sharing a meal, as well as by helping food retailers to adapt to changing consumption habits. However, there is a lack of systematic analysis of the quantitative evaluation of these campaigns. In particular, it is unclear which segment of the population is sensitive to such a behavioral campaign. The only published evaluation that we are aware of is qualitative [37] and does not provide data-driven guidelines for a large-scale implementation. Public and private health and environmental agencies would greatly benefit from a better understanding of the individual factors involved in the Meatless Monday commitment by the public. Such information would inform targeted public health communication campaigns about reducing meat consumption and promoting nutritionally adequate alternatives. The aim of the current study was to assess the immediate and short-term correlates of the campaign launched in France in January 2019 and promote weekly substitution of meat and fish (called “Green Monday”).

## 2. Hypothesis

First, we hypothesized that the target behavior, that is, choosing a vegetarian diet every Monday, would be especially endorsed among the socio-demographic, personality and attitudinal profiles that already have a lower motivational and behavioral opposition toward this change. Meat is seen as a typical male food [38,39,40,41,42] and, in Western countries, meat consumption is inversely related to age, educational level and socioeconomic status [43,44,45]. Therefore, a higher participation of females, younger and more educated and affluent individuals was expected. Social influences also represent significant determinants of food choices and dietary changes. An increasing number of studies demonstrated a network influences on food habits [46] and indicated that beyond similarity [47], couples and friends shape one another’s choices [48,49]. We therefore expected that the mere number of vegetarians in one’s network would increase the commitment toward Green Monday.

Personality factors also represent important predictive dimensions of environmental [50] and health behavior [51,52,53,54,55] as well as dietary habits [56,57]. Personality research in the last 30 years has been largely based on the Big Five dimensions [58] labeled Openness, Conscientiousness, Extraversion, Emotional stability and Agreeableness. The Big Five is a parsimonious taxonomy that is not dependent on factor analytical techniques [59] or assessment methods [60] and is considered as a cross-culturally reliable and a universal taxonomy [61]. Longitudinal studies showed that personality has a significant influence on human values, attitudes and human choices [62]. Regarding dietary patterns, the Openness dimension appears to be especially relevant since it captures “an individual’s cognitive flexibility, need for variety, and depth of emotional experience” [63]. Like the other dimensions, Openness is strongly heritable [64] and very stable across adult lifespan [65]. This personality dimension represents a constant correlate of IQ [66] and appears as the most consistent correlate of healthy habits [57,67,68,69,70] and willingness to try new foods [71] and to endorse pro-environmental behavioral intentions and behaviors [50,72,73,74,75,76]. People with high Openness (self-rated and, in some studies, informant-rated) consume more vegetables, fruits and cereals, and are less likely to consume meat products frequently, while controlling for age, gender, and educational level [43,56,57,70,75,77,78,79,80,81,82]. This link may be explained by the normative status endorsed by meat in most of the cultures. Meat represents the most traditional diet in Western societies [83,84], and any deviation from this normative diet implies more intellectual curiosity, flexibility and innovation. We therefore expected a positive relationship between Openness and the commitment to the Green Monday in France.

Beyond the comparison of participants signing in for Green Monday with a national sample, we also investigated the contribution of every variable exposed above on the statistical prediction of the intensity of participation, which was estimated by the number of weeks participants indicated they put Green Monday into practice from the first to the fifteenth week after the launch of the national campaign.

## 3. Methods

### 3.1. Features of the Campaign

The campaign, called “Green Monday” (or “Lundi Vert” in French), officially started in France on 7 January 7 2019 with a massive press release of the petition with 500 signatures of public figures including artists, sportsmen, politicians, scientists and NGOs calling on consumers to change their eating habits and avoid eating meat and fish every Monday throughout 2019 for environmental, health, and animal welfare considerations. The newspaper Le Monde published the petition, and the news was also spread by a global news agency (AFP). Most of the French news websites, as well as public television and radio broadcast, echoed the campaign. The campaign also included large 3 × 4 posters displayed in 60 subway stations in Paris. People belonging to the GM sample (see below) were invited to commit by filling in a questionnaire on a dedicated website (www.lundi-vert.fr). They filled in the questionnaire in January 2019. In accordance with research showing that monitoring and providing feedbacks to participants promotes behavioral change [85], we informed participants that they would receive weekly reminders (every Monday) during one whole year.

### 3.2. Measures

Participants in both samples were asked to report their city size (from 1 = less than 10,000 habitants to 6 = more than 400,000 habitants), their educational level (from 1 = lowest level, below baccalaureate [86]; 2 = Baccalaureat; 3 = 2 year-degree after baccalaureate; 4 = Bachelor’s degree; 5 = Master (first year); 6 = Master (2nd year); 7 = Doctorate or other degrees), their current affluence (from 1 = not well off at all to 5 = very well off). Two separate food frequency questions were used to investigate the importance of meat and fish consumption on ordinary weeks (from 1 = No, usually I don’t eat meat/fish to 5 = Yes, I eat meat/fish every day of the week). We also asked participants to indicate the number of vegetarians they counted in their family and among their friends (from 0 = Nobody to 5 = Five people or more). We also included a short version of Openness. Participants were invited to describe themselves on the traits such as “creative” or “imaginative” (1 = totally wrong; 5 = totally right). The whole scale was based on 6 items, Cronbach’s Alpha = 0.749) [87].

No other personality dimensions from the Big Five Model were included in the survey, but there were additional items on food preferences and representations which are not reported here. Missing data or the use of the non-response option (<2% per variable) were imputed using a straightforward linear interpolation method. Participants who did not indicate their gender (*n* = 38, 0.14%) were not included in the analysis.

### 3.3. National Comparison Sample (NC)

A national comparison sample (NC) was simultaneously recruited in order to be contrasted to the Green Monday sample. The NC participants were recruited by the Toluna Panel, a market research institute operating in more than 60 countries worldwide and involved in various epidemiological studies [88]. The panels are representative of the general population with respect to age, education, gender and regions. Participants had agreed to participate in online research and were screened for the following eligibility criteria: aged 18 years or older and French speaking. Respondents received points for completing the survey through Toluna, which they could redeem for rewards. The following questions were included in the NC survey: Have you heard about the Green Monday (without meat and fish)? If so, do you intend to participate? 1 = Yes, but I have not started yet, 2 = Yes, and I have already started, 3 = No, 4 = I don’t answer.

### 3.4. Green Monday Sample (GM)

At the very end of the questionnaire, participants were asked the following question: Do you agree to replace meat and fish every Monday? With the following answers: 1 = Yes, 2 = No, 3 = I’m already doing it. Accordingly, participants in the GM sample received two emails every week. On every Monday, they received a reminder and some vegetarian recipes, and then on Tuesday they were sent an email asking them to merely indicate if they had (coded 1) or not (coded 0) put the Green Monday into practice the previous day. In this study, we analyzed the binary replies of participants between the first week after the campaign launch (Monday 7 January, 2019) and the fifteenth week.

### 3.5. Statistical Analyses

We first compared participants in the Green Monday sample (GM) to the National Comparison sample (NC) based on Chi-squares and T-tests applying Bonferroni corrections. Then, a logistic regression analysis was performed to estimate Odds Ratio (OR) and 95% Confidence Interval (95% CI) of every variable to predict the belonging to the GM sample (coded 2) by contrast with the NC sample (coded 1). Age and gender were entered into block 1 of a multivariate analysis. In block 2, each potential predictive factor was added stepwise to the model by using an automated forward selection procedure. The significance level to select variables that remained in the model was *p* < 0.05. Finally, in the follow-up study, we carried out a linear regression in order to analyze the specific contribution of every variable to the length of the practice of Green Monday. Age and sex were entered into block 1 of the multivariate analysis. In block 2, each potential predictive factor was added stepwise to the model by using an automated forward selection procedure. The significance level to select variables that remained in the model was *p* < 0.05.

## 4. Results

### 4.1. Preliminary Analysis

In February 2019, one month after the campaign launch, 24,507 participants had filled in the Green Monday questionnaire. Among the respondents, 86.1% answered that they had agreed to replace meat and fish every Monday, 13.1% indicated that this was already the case, and 0.7% declined to commit. In the following analysis, we exclusively focus on the 86.1% participants who had decided to begin the Green Monday. The final Green Monday sample (GM) included 21,112 participants and was compared to a National Comparison sample (NC) of 2005 participants (Table 1) sampled the first week of February 2019.

In the NC sample, 51.2% of the respondents said they had heard about Green Monday. Using Chi-squares and t-tests (applying Bonferroni correction), we compared the participants who had heard and those who had not heard about the French Green Monday campaign. No significant differences emerged in any variables. Lastly, in the national sample, 10.5% indicated that they had already started to apply the Green Monday, and 25.1% intended to do so.

As the univariate analysis indicated (Table 1), participants belonging to the Green Monday sample were more often females, were younger, lived in larger cities, had a higher educational levels and higher affluence, consumed meat and fish less frequently, counted more vegetarians in their social network, and scored higher on the Openness scale.

### 4.2. Main Analysis

The analysis (Table 2) showed that beyond age and gender, participants belonging to the Green Monday sample were more educated and had higher self-rated affluence. In this sample, the size of participants’ vegetarian network was greater, the frequency of meat consumption was slightly higher, while the frequency of fish consumption was lower. Finally, the personality trait Openness was more strongly endorsed by participants in GM sample. The overall model accounted for 22% of the variance (Nagelkerke pseudo R^2^) [89].

### 4.3. Follow up Study

The length of the practice was then analyzed in the GM sample. Every Tuesday, participants were sent a participation email. The response rate per week ranged between 52.8% (*n* = 11,137) and 70.5% (*n* = 14,884). Focusing only on non-missing values, we summed up the number of weeks participants declared they had practiced Green Monday (practiced = 1; did not practice = 0). The total sample answering 15 times was *n* = 5165. The minimum length was 0 week while the maximum was 15 weeks (M = 12.51, SD = 2.62). The characteristics of the all the participants, as well as the characteristics of those who answered every 15 weeks and those who did not answer at any point [90], are reported in Table 3.

The analysis showed that the more participants used to eat meat before the beginning of the campaign, the shorter time they applied the Green Monday (beta = −0.10, t = −7.48, *p* < 0.000). Moreover, the number of weeks participants practiced Green Monday was lower among participants with higher educational level (beta = −0.05, t = −3.72, *p* < 0.000) and who lived in larger cities (beta = −0.04, t = −2.85, *p* < 0.004) but was higher among older participants (beta = 0.16, t = 11.35, *p* < 0.000). Finally, participants with higher Openness applied Green Monday longer than the others (beta = 0.03, t = 2.62, *p* < 0.009). Participants’ gender or affluence, the number of vegetarians in their social network and their weekly consumption of fish were not significantly associated with the length of their commitment. The value of the adjusted R^2^ of the final model was 0.056.

In order to rule out the possibility that the older people (aged over 65 years) may not willing to eat meat more than younger people do by aging, we also analyzed the data without participants aged over 65 and confirmed the inverse effect of age on the length of the Green Monday practice. The results remained almost identical regarding the contribution of age (beta = 0.13, *p* < 0.000).

## 5. Discussion

To the best of our knowledge, this study represented the first systematic attempt to evaluate a national meatless day campaign. Immediately after the launch in France in January 2019, and fifteen weeks later, the responsiveness toward the campaign was significantly higher in specific segments of the population. Compared to an independent national comparison sample, people who enrolled in Green Monday were more frequently females, younger, more affluent participants with a higher education, and lived in bigger cities. As hypothesized, the trait Openness was higher in the Green Monday sample compared to the National comparison sample and was also related to the number of participation weeks in the GM sample, which may reflect the preference for cultural innovation frequently conveyed by this personality trait.

The analysis showing that the Green Monday attracted specific profiles may usefully inform further meat reduction campaigns. While the mere existence of these features may be seen as factors hindering the wide introduction of dietary changes, they may also be taken into account to shape persuasive communication messages, and more precisely tailored communication [91,92,93]. For example, one may speculate that in order to reach individuals with a low level of Openness, it may be appropriate to remind them that nourishing vegetables, such as kidney beans, lentils or chickpeas are part of “traditional food”. Increasing the familiarity and the cultural legitimacy of leguminous plants may help people with lower Openness to consider Green Monday initiatives. Following the studies showing that congruency between environmental appeals and political values stimulates pro-environmental behavior [94,95], the design of future campaigns may thus be strengthened by including individual factors.

Finally, five limitations of the current research should be mentioned. First, all of the analyzed variables were self-reported, which has inherent limitations when it comes to measuring behavior and behavioral change [96,97]. Second, the non-response rate observed in the follow-up analysis we carried out in the GM sample was substantial: every week, between 25% and 50% of the participants did not reply to the email they were sent. Only participants who kept on answering every week were included in the follow-up study, which therefore represents only a small fraction of the total sample when all the binary replies of the 15 weeks are aggregated. Third, this study did not have information on participants who had diseases, which may change their dietary habits such as diabetes, hypertension, cardiovascular disease, and cancer. Fourth, the generalization of the results and their usefulness for diet change at a global level should also take into account that reducing the frequency of meat eating represents a specific strategy that may appeal to a segment of consumers that is not equivalent to the strategy which consists in reducing meat portion size [98]. Finally, we did not measure the quantity of meat participants consumed the other days of the week. A “rebound effect” of extra meat consumption after a meatless day cannot be excluded [98].

In conclusion, despite the limitations we mentioned, the current study provided original insights that shed light, for the first time, on the immediate and short-term individual correlates of a national campaign promoting weekly substitution of meat and fish by vegetables. We showed that a personality dimension was an important facilitator to enroll and to sustain the enrollment decision: participants with a higher level of the trait Openness signed in more frequently for the Green Monday initiative and put their decision in action during a longer period of time. Such results could inform targeted public health communication campaigns about reducing meat consumption and promoting healthier and more sustainable diets in the public.

## Figures and Tables

**Table 1 nutrients-11-01694-t001:** Characteristics of the two samples and univariate comparisons.

	Green Monday	National Comparison	Statistical Tests
	(*n* = 21,112)	(*n* = 2005)	
Gender			
Women (%)	77.5	47.7	*χ*^2cor^(1) = 864.117 *p* < 0.000
Age (%)			
18–25	2.4	9.0	*χ*^2^(16) = 542.100 *p* < 0.000
26–30	11.6	7.8	
31–35	13.2	8.1	
36–40	12.2	9.1	
41–45	12.0	9.4	
46–50	11.0	9.9	
51–55	10.4	9.4	
56–60	9.4	8.9	
61–65	7.2	9.2	
66–70	5.5	12.2	
71–75	3.2	5.7	
76–80	1.5	0.9	
81–85	0.3	0.3	
86–90	0.1	0.0	
91–95	0.0	0.0	
96–100 or more	0.0	0.0	
City size (%)			
Less than 10,000 inhabitants	40.5	41.6	*χ*^2^(5) = 121.096, *p* < 0.000
From 10,000 to 20,000 inhabitants	10.6	13.1	
From 20,000 to 50,000 inhabitants	12.8	14.7	
From 50,000 to 100,000 inhabitants	8.3	11.9	
From 100,000 to 400,000 inhabitants	11.5	10.2	
More than 400,000 inhabitants	16.4	8.4	
Education (%)			
Less than baccalaureate	9.2	23.8	*χ*^2^(6) = 846.07, *p* > 0.000
Baccalaureate	14.4	24.9	
2 years-degree after baccalaureate	16.5	21.4	
Bachelor degree	14.9	10.5	
Master (first year)	9.8	6.2	
Master (2nd year)	28.9	11.0	
Doctorate or other degrees	6.3	2.1	
Frequency of meat consumption (%)			
Every day	16.0	15.1	*χ*^2^(4) = 56.76, *p* > 0.000
5 or 6 days a week	23.1	22.6	
3 or 4 days a week	31.1	35.1	
1 or 2 days a week	21.3	23.1	
Normally, I don’t consume meat	8.5	4.1	
Frequency of fish consumption	(%)		
Every day	1.4	1.2	*χ*^2^(4) = 46.78, *p* > 0.000
5 or 6 days a week	1.2	2.3	
3 or 4 days a week	9.7	10.9	
1 or 2 days a week	72.0	74.5	
Normally, I don’t consume fish	15.7	11.1	
Affluence (1 to 5) (Mean, SD)	3.33 (0.93)	2.85 (1.02)	tcor (2332.10) = 20.23, *p* < 0.000
Veget. Network (0 to 5+) (Mean, SD)	0.99 (1.38)	0.40 (0.88)	tcor (3037.91) = 26.91, *p* < 0.000
Openness (6 items, 1 to 5) (Mean, SD)	3.65 (0.60)	3.36 (0.69)	tcor(2298.69) = 18.49, *p* < 0.000

Statistical tests were chi squares (*χ*^2^) and t tests with Bonferroni corrections (*tcor*) when relevant.

**Table 2 nutrients-11-01694-t002:** Multivariate Logistic Regression predicting the belonging to Green Monday (GM) sample (coded 2) by contrast with the National Comparison (NC) sample (coded 1).

	Odds Ratio	CI	*p*
Age	1.00	0.96–1.00	0.096
Gender	4.26	3.86–4.71	0.000
City size	0.96	0.93–0.99	0.019
Educational level	1.32	1.28–1.36	0.000
Self-rated affluence	1.54	1.42–1.58	0.000
Meat consumption (frequency)	1.05	1.01–1.10	0.010
Fish consumption (frequency)	0.81	0.76–0.87	0.000
Size of vegetarian network	1.50	1.41–1.58	0.000
Openness	1.79	1.65–1.93	0.000

**Table 3 nutrients-11-01694-t003:** Characteristics of all participants, of those that answered all 15 weeks and those lost at any follow up week.

	All Participants	Remaining Participants	Participants Lost to Follow-Up
	(*n* = 21,112)	(*n* = 5165)	(*n* = 15,947)
Gender			
Women (%)	77.5	77.6	77.5
Age (%)			
18–25	2.4	1.0	2.9
26–30	11.6	8.4	12.7
31–35	13.2	12.1	13.6
36–40	12.2	12.4	12.1
41–45	12.0	11.4	12.1
46–50	11.0	11.5	10.8
51–55	10.4	10.7	10.3
56–60	9.4	10.5	9.0
61–65	7.2	8.2	6.8
66–70	5.5	7.2	5.0
71–75	3.2	4.1	2.9
76–80	1.5	2.0	1.3
81–85	0.3	0.3	0.3
86–90	0.1	0.1	0.1
91–95	0.0	0.0	0.0
96–100 or more	0.0	0.0	0.0
City size (%)			
Less than 10,000 inhabitants	40.2	39.2	40.5
From 10,000 to 20,000 inhabitants	10.7	10.9	10.6
From 20,000 to 50,000 inhabitants	13.0	13.9	12.7
From 50,000 to 100,000 inhabitants	8.5	8.6	8.5
From 100,000 to 400,000 inhabitants	11.5	12.0	11.3
More than 400,000 inhabitants	16.2	15.5	16.5
Education (%)			
Less than baccalaureate	9.2	8.1	9.6
Baccalaureate	14.4	12.6	15.0
2 years-degree after baccalaureate	16.5	16.2	16.6
Bachelor degree	14.9	14.5	15.0
Master (first year)	9.8	10.6	9.5
Master (2nd year)	28.9	30.8	28.5
Doctorate or other degrees	6.3	7.1	6.0
Frequency of meat consumption (%)			
Every day	16.0	15.1	16.4
5 or 6 days a week	23.1	23.8	22.9
3 or 4 days a week	31.1	31.0	31.1
1 or 2 days a week	21.3	21.0	21.4
Normally, I don’t consume meat	8.5	9.1	8.4
Frequency of fish consumption (%)			
Every day	1.4	1.1	1.5
5 or 6 days a week	1.2	1.2	1.2
3 or 4 days a week	9.7	10.4	9.5
1 or 2 days a week	72.0	73.4	71.6
Normally, I don’t consume fish	15.7	13.9	16.2
Affluence (1 to 5) (Mean, SD)	3.33 (0.93)	3.42 (0.88)	3.30 (0.94)
Veget. Network (0 to 5+) (Mean, SD)	0.99 (1.38)	0.97 (1.34)	1.00 (1.40)
Openness (6 items, 1 to 5) (Mean, SD)	3.65 (0.60)	3.61 (0.60)	3.67 (0.59)

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
