# Peer review of "Immediate and 15-Week Correlates of Individual Commitment to a “Green Monday” National Campaign Fostering Weekly Substitution of Meat and Fish by Other Nutrients"

_nutrients, 2019, doi:10.3390/nu11071694_

Reviewer 1 Report

General overview

This manuscript addresses a very important research topic given current and future health and sustainability issues associated with meat consumption. It is important to build evidence on initiatives and interventions to reduce meat consumption such as the Meatless Monday campaign. The manuscript presents an overall clear and focused contribution, drawing on two large samples that are relevant to address the main aims of the study. It is also generally well grounded in terms of the analytical procedures it uses, and the conclusions drawn from the data. In spite of these strengths, I have a small but relevant set of comments and suggestions that ought to be addressed.

Comments and suggestions

1. In the Methods, please: (1) split this section into smaller subsections to enable a more focused reading; (2) provide more details about the Openness measure (e.g., question and examples of items); (3) if possible, provide more details about the communication campaign that launched the “Lundi Vert” initiative, and the components delivered/included in the initiative; and (4) change ‘1=lowest level’ of education to present the actual level of education that value 1 refers to.

2. In the Results section: (1) preliminary analysis – please include information (in text) about the statistical analyses that were performed; (2) follow-up study – as supplement, to enable a more focused reading and interpretation, please include a table/comparison with the baseline characteristics (gender, age, city size, education, frequency of meat consumption, frequency of fish consumption, affluence, vegetarian network, openness) considering: (i) all MM participants, (ii) MM participants lost to follow-up, and (iii) remaining MM participants [for an example see Dumville, J. C., Torgerson, D. J., & Hewitt, C. E. (2006). Reporting attrition in randomised controlled trials. BMJ, 332(7547), 969-971].

3. This is only a detail and the manuscript is clear and well-written, but check the document for missing plural and small mishaps/mistakes (e.g., p.3, l.116 – will/would; p.3, l.120 – question/questions; p.4, l.157 – difference/differences).

Author Response

Response to Reviewer 1 Comments

 First of all, my co-author and myself would like to thank you for the time and the expertise you devoted to our manuscript  « Immediate and 15 weeks Impact Following a National Campaign Promoting Weekly Substitution of Meat and Fish». We appreciated your very stimulating comments : « This manuscript addresses a very important research topic given current and future health and sustainability issues associated with meat consumption. It is important to build evidence on initiatives and interventions to reduce meat consumption such as the Meatless Monday campaign. The manuscript presents an overall clear and focused contribution, drawing on two large samples that are relevant to address the main aims of the study. It is also generally well grounded in terms of the analytical procedures it uses, and the conclusions drawn from the data ».

 We also acknowledge than the manuscript could be significantly improved following your advices. Please find below our replies to your comments.

1.In the Methods, please: (1) split this section into smaller subsections to enable a more focused reading; (2) provide more details about the Openness measure (e.g., question and examples of items); (3) if possible, provide more details about the communication campaign that launched the “Lundi Vert” initiative, and the components delivered/included in the initiative; and (4) change ‘1=lowest level’ of education to present the actual level of education that value 1 refers to.

(1)The Methods section is now split into smaller subsections

(2) Sample items of Openness dimension are now included

(3) We added a few more details about the communication campaign

(4) We specified what educational levels means

 2. In the Results section: (1) preliminary analysis – please include information (in text) about the statistical analyses that were performed; (2) follow-up study – as supplement, to enable a more focused reading and interpretation, please include a table/comparison with the baseline characteristics (gender, age, city size, education, frequency of meat consumption, frequency of fish consumption, affluence, vegetarian network, openness) considering: (i) all MM participants, (ii) MM participants lost to follow-up, and (iii) remaining MM participants [for an example see Dumville, J. C., Torgerson, D. J., & Hewitt, C. E. (2006). Reporting attrition in randomised controlled trials. BMJ, 332(7547), 969-971].

(1) We indicated information about the statistical analysis that we performed

(2) The requested table is added in the revision following the model of Dumville et al. (2006)

 3. This is only a detail and the manuscript is clear and well-written, but check the document for missing plural and small mishaps/mistakes (e.g., p.3, l.116 – will/would; p.3, l.120 – question/questions; p.4, l.157 – difference/differences).

 The mistakes are corrected in the revised version.

 Reviewer 2 Report

This research article concludes that the impact of the national campaign was higher among specific demographic and personality profiles. The research theme is interesting and important.

However, there are several things need to reconsider. With these methods, it may be difficult to suggest the effect of the Meatless Monday campaign. The followings are points the author needs to reconsider.

Abstract:

Major points

1. Although there is a limited word count, you should include the statistical methods.

Minor points

2. Line 16-19: You should explain what these results (values) were. I suspect these results are ORs and 95% CI, and please indicate the abbreviation with full words.

3. Line 23: In my opinion, the result of “t = 2.65” is not much necessary.

Introduction

Major points

1. I understand the environmental issues such as greenhouse gas emission is one of the important issues. However, this journal is for nutrients (or food for health). Please describe more about the health issues on excessed meat consumption than environmental issues in addition to line 33-34.

2. Please introduce the national guideline of the amount of meat (or protein). In addition, please indicate the current situation of the percentages of the population who over-consumed meat (or protein).

 Methods

Major points

1. There was no description of statistical analyses. Please explain in the detail.

2. It may be difficult to suggest the effect of the campaign with comparing the characteristics of participants between Meatless Monday in this study and Green Monday in national comparison sample or with investigating the association of sociodemographic items with the length of the practice of Meatless Monday. The effect should be, for example, the difference of health indices, such as physical measurements or mental health, between before and after the practice.

3. The older people (aged over 65 years) may not willing to eat meat more than younger people do by aging. Please indicate whether the results did not change even excluding older people.

If you have information, please indicate whether the participant had diseases which have a possibility to change their dietary habits such as diabetes, hypertension, cardiovascular disease, or cancer.

4. Please indicate the ethical statement.

 Results

Major points

1. According to the standard style of epidemiology, we usually do not describe the statistical analyses in the method section. Please make a "Statistical analyses" in the Method section.

2. Please indicate a table of results of the main analyses as Table 2.

 Minor points

1. Line 139; "NS" may need be corrected "NC". Please check.

2. Line 125-126: If you did not have a particular reason, you do not have to describe personality detentions you did not use to be simpler.

3. Table 1: Please explain the statistical analyses you use in the footnote. Please show what you indicated using with x2(**) and tcor (**)?

4. Table 1 the percentage of 26-30 years indicated in the different column from others.

 Conclusion (it may better to change the tile to “Discussion”)

1. Line 262-269: It may not an effect of the Meatless Monday campaign. It seems this article clarified the characteristics of the participants of this campaign.

2. Line 271-280: I feel the discussion in this paragraph is only a speculation. Please add previous studies to indicate what your discussion came from.

Author Response

Response to Reviewer 2 Comments

 First of all, my co-author and myself would like to thank you for the time and the expertise you devoted to our manuscript  « Immediate and 15 weeks Impact Following a National Campaign Promoting Weekly Substitution of Meat and Fish». We appreciated your very stimulating comments : « This research article concludes that the impact of the national campaign was higher among specific demographic and personality profiles. The research theme is interesting and important».

 We also acknowledge than the manuscript could be significantly improved following your advices. Please find below our replies to your comments.

Abstract: Major points

1. Although there is a limited word count, you should include the statistical methods.

-The statistical methods is now included in the abstract

Abstract: Minor points

2. Line 16-19: You should explain what these results (values) were. I suspect these results are ORs and 95% CI, and please indicate the abbreviation with full words.

We did the requested changes

3. Line 23: In my opinion, the result of “t = 2.65” is not much necessary.

 This information is deleted in the revised version.

Introduction : Major points

1.I understand the environmental issues such as greenhouse gas emission is one of the important issues. However, this journal is for nutrients (or food for health). Please describe more about the health issues on excessed meat consumption than environmental issues in addition to line 33-34.

New contents related to health are added in the revised version.

2. Please introduce the national guideline of the amount of meat (or protein). In addition, please indicate the current situation of the percentages of the population who over-consumed meat (or protein).

Both requested information are added in the revised version

Methods : Major points

1. There was no description of statistical analyses. Please explain in the detail.

2. It may be difficult to suggest the effect of the campaign with comparing the characteristics of participants between Meatless Monday in this study and Green Monday in national comparison sample or with investigating the association of sociodemographic items with the length of the practice of Meatless Monday. The effect should be, for example, the difference of health indices, such as physical measurements or mental health, between before and after the practice.

Unfortunately, this information is not available.

3. The older people (aged over 65 years) may not willing to eat meat more than younger people do by aging. Please indicate whether the results did not change even excluding older people.

We agree that motivation to limit meat consumption may depend on participant’s age. In all of our analysis, we therefore controlled for age in logistic regression (comparison between both samples) and in multiple regression analysis (in order to predict the number of weeks participants put green Monday into practice). In order to follow your suggestions and check for the stability of the results, we also analyzed the data without participants aged over 65 and confirmed the inverse effect of âge on the lenght of Green Monday practice. The results remained almost identical (beta = .13, p.<.000 instead of beta = .16, p.<.000).< span="">

If you have information, please indicate whether the participant had diseases which have a possibility to change their dietary habits such as diabetes, hypertension, cardiovascular disease, or cancer

Unfortunately, this information is not available

4. Please indicate the ethical statement.

 The ethical statement is included in the revised version.

Results : Major points

1.According to the standard style of epidemiology, we usually do not describe the statistical analyses in the method section. Please make a "Statistical analyses" in the Method section.

In our manuscript, there is a « Methods » section and a « Results » section. The statistical analyses are located in the results section.

2. Please indicate a table of results of the main analyses as Table 2.

 We added the requested table in the revised version of the manuscript

Minor points

1.Line 139; "NS" may need be corrected "NC". Please check.

The mistake is corrected.

2. Line 125-126: If you did not have a particular reason, you do not have to describe personality detentions you did not use to be simpler.

We understand that it may seems unuseful to mention personality dimensions that we did not use. However, in many studies, the 5 factors are measured simultaneously, and we believe that it may be important to indicated that we did not hide some results.

3. Table 1: Please explain the statistical analyses you use in the footnote. Please show what you indicated using with x2(**) and tcor (**)?

This point is clarifed in the new version.

4. Table 1 the percentage of 26-30 years indicated in the different column from others.

This issue is corrected in the new version.

Conclusion (it may better to change the title to “Discussion”)

In the revised version, the title is now « Discussion »

1.Line 262-269: It may not an effect of the Meatless Monday campaign. It seems this article clarified the characteristics of the participants of this campaign.

We removed the idea of causality to underline the descriptive feature of our study.

2. Line 271-280: I feel the discussion in this paragraph is only a speculation. Please add previous studies to indicate what your discussion came from.

Three new references are added on tailored communication.

Round  2

Reviewer 1 Report

Thank you for the effort and care put into the preparation of the revised version of the manuscript. The comments and concerns raised in the reviews were addressed, which in my view strengthened the contribution that the manuscript offers the field.

Author Response

Thank you very much for the time and expertise you devoted to our article.

All the best.

Reviewer 2 Report

Dear author

Thank you for considering the revision. Some parts became clear but some did not.

Please excuse for pointing additional things out. I found some minor errors, please recheck precisely in the manuscript.

 Overall comment

As I suggested before in the Discussion, this study cannot describe an impact (or effect) on the Meatless Monday campaign. Possible quantitative evaluative designs for estimating the magnitude of the benefits on non-experimental designs were majorly recommended as follows: "uncontrolled before and after", "controlled before and after", and "time series analysis" (Eccles M et al. Qual Saf Health Care. 2003). I understand this article clarified the characteristics of the participants of this campaign but did not show a magnitude of the effects on the GM campaign because you did not compare indicators before and after the GM.

You may better to change the expression in the title and manuscript from “Impact” to “characteristics” or another proper word.

 Abstract:

2. Line 16-19: You should explain what these results (values) were. I suspect these results are ORs and 95% CI, and please indicate the abbreviation with full words.

 >Ans. We did the requested changes

 >>Response. Line 21: Please revise as follows.

Odds Ratio (OR)= 0.23 (95% Confidence Interval [CI]: 0.21-0.25)

 Introduction

Additional comment 1. Line 39-40: According to the reference you employed, it is better to describe “…excess meat consumption such as risks of cancer13, heart diseases14 15, strokes16 17 18, type 2 diabetes19, and obesity20 21”.

Line 39: Reference 14 is about mortality and should delete because all of the other references are about the risk of diseases. Please check.

 Methods:

3. The older people (aged over 65 years) may not willing to eat meat more than younger people do by aging. Please indicate whether the results did not change even excluding older people.

 >Ans. We agree that motivation to limit meat consumption may depend on participant’s age. In all of our analysis, we therefore controlled for age in logistic regression (comparison between both samples) and in multiple regression analysis (in order to predict the number of weeks participants put green Monday into practice). In order to follow your suggestions and check for the stability of the results, we also analyzed the data without participants aged over 65 and confirmed the inverse effect of age on the lenght of Green Monday practice. The results remained almost identical (beta = .13, p.<.000 instead of beta = .16, p.<.000).< p="">

 >>Response: Please describe the result you showed above in the Method and Result section as additional analysis and result.

 4. If you have information, please indicate whether the participant had diseases which have a possibility to change their dietary habits such as diabetes, hypertension, cardiovascular disease, or cancer.

 >Ans. Unfortunately, this information is not available

 >>Response: Please add above those as one of the limitations in the discussion; this study did not have information of participants who had diseases which have a possibility to change their dietary habits such as diabetes, hypertension, cardiovascular disease, and cancer.

 Additional comment 1. Line 137-155: I could not find when this survey was performed (i.e. Before initiating GM). Please explain.

 Results:

1.According to the standard style of epidemiology, we usually do not describe the statistical analyses in the method section. Please make a "Statistical analyses" in the Method section.

 >Ans. In our manuscript, there is a « Methods » section and a « Results » section. The statistical analyses are located in the results section.

 >>Response: Please follow the STROBE statement as long as possible (https://www.strobe-statement.org/index.php?id=available-checklists). Statistical analysis should be included in the Method section and indicate only results in the Result section. If you have a specific reason, please explain.

 Minor points

3. Table 1: Please explain the statistical analyses you use in the footnote. Please show what you indicated using with x2(**) and tcor (**)?

 >Ans. This point is clarifed in the new version.

 >> Response and Additional comment: Please add more detail in the footnote in addition to “Bonferroni correction” so that we can understand what results showed in the tables.

-1. There is "reference 100" in the title in Table 1. Please delete if it is not necessary.

-2. There is one line space between “Gender” and “Women”.

-3. It is still unclear what parenthesis indicated in x2 (**) and tcor (**). Is this the number for Bonferroni correction? Please define in the manuscript and Table 1.

-4. Related to -3, please define what x2 and tcor indicate, for example, x2 is the chi-square test and tcor is the Pearson correlation coefficient

-5. Did the “,” which was the values of x2 indicate a decimal point or apostrophe for 1,000? Please check.

-6. About the first line, “x2cor(2)=864,117”, are there any reasons to indicate “cor” for only this result?

-7. There are about 2-3 line spaces at the bottom of Table 1.

-8. Please indicate full words of GM and NC to be able to understand results without reconfirming the manuscript.

-9. In Table2, I recommend OR, CI, and p-value in order. I think “B” is less important if OR and 95%CI is indicated.

-10. You might set the analysis that “NC” indicates the event happened. However, according to your description, you want to show the event of “GM participants”. It is better to change the setting to “GM” as an event happened (e.g., OR of female may change from 0.23 to 1.xx).

-11. It is unclear what variable you employed as a reference to show ORs if the variable was a category. I introduce an article just for an example of the table for the result (https://www.ncbi.nlm.nih.gov/pmc/articles/PMC4641714/pdf/pone.0142779.pdf). Please specify the reference if you used categorical variables.

-12. Line 263-273: It is better to describe not only the significant association but also how variables associated with GM participants. The way of description in the Abstract in line 19-26 is better than that in the manuscript in line 263-273.

-13. In Table3, line 315, what is “1” in the participants in the column of “lost of follow-up”?

-14. The form of table lines in Table1, 2, and 3 was not unified. For example, there is no line above the title of items in Table 1 and 3. Only Table2 had two lines for the title of the value.

Author Response

Dear reviewer,

Please find enclosed our replies to your comments and suggestions. 

We thank you very much for the time and expertise you devoted to our paper and hope that our new version will meet your expectations. 

Our replies are written in a blue font.

-------

Overall comment

As I suggested before in the Discussion, this study cannot describe an impact (or effect) on the Meatless Monday campaign. Possible quantitative evaluative designs for estimating the magnitude of the benefits on non-experimental designs were majorly recommended as follows: "uncontrolled before and after", "controlled before and after", and "time series analysis" (Eccles M et al. Qual Saf Health Care. 2003). I understand this article clarified the characteristics of the participants of this campaign but did not show a magnitude of the effects on the GM campaign because you did not compare indicators before and after the GM. You may better to change the expression in the title and manuscript from “Impact” to “characteristics” or another proper word.

We agree that strictly speaking, « impact » is too strong. We elected to replace the title into “Immediate and 15-weeks Correlates of Individual Commitment to a National Campaign “Green Monday” Fostering Weekly Substitution of Meat and Fish by other nutrients ». Moreover, we carefully replaced every mention of an impact by another expression that does not imply causality.

Abstract:

2. Line 16-19: You should explain what these results (values) were. I suspect these results are ORs and 95% CI, and please indicate the abbreviation with full words.

 >Ans. We did the requested changes

 >>Response. Line 21: Please revise as follows.

Odds Ratio (OR)= 0.23 (95% Confidence Interval [CI]: 0.21-0.25)

 We did the requested change in the new revision.

Introduction

Additional comment 1. Line 39-40: According to the reference you employed, it is better to describe “…excess meat consumption such as risks of cancer13, heart diseases14 15, strokes16 17 18, type 2 diabetes19, and obesity20 21”.

Line 39: Reference 14 is about mortality and should delete because all of the other references are about the risk of diseases. Please check.

 All the requested changes are applied in the new revision

Methods:

3. The older people (aged over 65 years) may not willing to eat meat more than younger people do by aging. Please indicate whether the results did not change even excluding older people.

 >Ans. We agree that motivation to limit meat consumption may depend on participant’s age. In all of our analysis, we therefore controlled for age in logistic regression (comparison between both samples) and in multiple regression analysis (in order to predict the number of weeks participants put green Monday into practice). In order to follow your suggestions and check for the stability of the results, we also analyzed the data without participants aged over 65 and confirmed the inverse effect of age on the lenght of Green Monday practice. The results remained almost identical (beta = .13, p.<.000 instead of beta = .16, p.<.000).< span="">

 >>Response: Please describe the result you showed above in the Method and Result section as additional analysis and result.

 This section is now pasted in the paper below the new title « Additional analysis ».

4. If you have information, please indicate whether the participant had diseases which have a possibility to change their dietary habits such as diabetes, hypertension, cardiovascular disease, or cancer.

>Ans. Unfortunately, this information is not available

>>Response: Please add above those as one of the limitations in the discussion; this study did not have information of participants who had diseases which have a possibility to change their dietary habits such as diabetes, hypertension, cardiovascular disease, and cancer.

This comment is added as requested in the discussion

 Additional comment 1. Line 137-155: I could not find when this survey was performed (i.e. Before initiating GM). Please explain.

This missing information is added .

Results:

1.According to the standard style of epidemiology, we usually do not describe the statistical analyses in the method section. Please make a "Statistical analyses" in the Method section.

 >Ans. In our manuscript, there is a « Methods » section and a « Results » section. The statistical analyses are located in the results section.

 >>Response: Please follow the STROBE statement as long as possible (https://www.strobe-statement.org/index.php?id=available-checklists). Statistical analysis should be included in the Method section and indicate only results in the Result section. If you have a specific reason, please explain.

In the new revision, this suggestion is taken into account. A new subtitle is introduced in the manuscript (Statistical analysis) with a description of the various analysis we performed.

 Minor points

3. Table 1: Please explain the statistical analyses you use in the footnote. Please show what you indicated using with x2(**) and tcor (**)?

>Ans. This point is clarifed in the new version.

>> Response and Additional comment: Please add more detail in the footnote in addition to “Bonferroni correction” so that we can understand what results showed in the tables.

-1. There is "reference 100" in the title in Table 1. Please delete if it is not necessary.

The reference is deleted.

-2. There is one line space between “Gender” and “Women”.

This space is suppressed.

-3. It is still unclear what parenthesis indicated in x2 (**) and tcor (**). Is this the number for Bonferroni correction? Please define in the manuscript and Table 1.

Full precisions are given in the table.

-4. Related to -3, please define what x2 and tcor indicate, for example, x2 is the chi-square test and tcor is the Pearson correlation coefficient

Full precisions regarding X2 and tcor are given in the table.

-5. Did the “,” which was the values of x2 indicate a decimal point or apostrophe for 1,000? Please check.

It is a decimal point. We checked the whole table.

-6. About the first line, “x2cor(2)=864,117”, are there any reasons to indicate “cor” for only this result?

Yes. (Yates correction is recommended for 4X4 tables using Chi square)

-7. There are about 2-3 line spaces at the bottom of Table 1.

The lines are suppressed.

-8. Please indicate full words of GM and NC to be able to understand results without reconfirming the manuscript.

We did the requested change in Table 1.

-9. In Table2, I recommend OR, CI, and p-value in order. I think “B” is less important if OR and 95%CI is indicated.

We did the requested changes.

-10. You might set the analysis that “NC” indicates the event happened. However, according to your description, you want to show the event of “GM participants”. It is better to change the setting to “GM” as an event happened (e.g., OR of female may change from 0.23 to 1.xx).

We followed this suggestion and reanalyzed the date after reversing the coding and changed the results accordingly in the paper.

-11. It is unclear what variable you employed as a reference to show ORs if the variable was a category. I introduce an article just for an example of the table for the result (https://www.ncbi.nlm.nih.gov/pmc/articles/PMC4641714/pdf/pone.0142779.pdf). Please specify the reference if you used categorical variables.

Our predictors are not categorical (except gender).

-12. Line 263-273: It is better to describe not only the significant association but also how variables associated with GM participants. The way of description in the Abstract in line 19-26 is better than that in the manuscript in line 263-273.

Actually, in the abstract, we described the results of the logistic regression while in the results section, we described the univariate tests and the results of the logistic regression.

As requested, in the revised version, we added statistical values corresponding to univariate analysis.

-13. In Table3, line 315, what is “1” in the participants in the column of “lost of follow-up”?

The « 1 » was a typo. We did the correction.

-14. The form of table lines in Table1, 2, and 3 was not unified. For example, there is no line above the title of items in Table 1 and 3. Only Table2 had two lines for the title of the value.

The features of the tables are standardized in the new revision.
